# Controlled Reduction of Graphene Oxide Using Sulfuric Acid

**DOI:** 10.3390/ma14010059

**Published:** 2020-12-25

**Authors:** Ana Cecilia Reynosa-Martínez, Erika Gómez-Chayres, Rafael Villaurrutia, Eddie López-Honorato

**Affiliations:** 1Centro de Investigación y de Estudios Avanzados del IPN, Unidad Saltillo, Av. Industria Metalúrgica 1062, Parque Industrial, Ramos Arizpe 25900, Coahuila, Mexico; cecilia.reynosa@cinvestav.edu.mx (A.C.R.-M.); erika.chayres@cinvestav.mx (E.G.-C.); 2Thermo Fisher Scientific de México, Avenida Morones Prieto 2805 Pte., Monterrey 64710, Nuevo León, Mexico; rafael.arenas@thermofisher.com; 3Oak Ridge National Laboratory, Oak Ridge, TN 37831, USA

**Keywords:** reduce graphene oxide, sulfuric acid, fuming sulfuric acid, UV-radiation

## Abstract

Sulfuric acid under different concentrations and with the addition of SO_3_ (fuming sulfuric acid) was studied as a reducing agent for the production of reduced graphene oxide (RGO). Three concentrations of sulfuric acid (1.5, 5, and 12 M), as well as 12 M with 30% SO_3_, were used. The reduction of graphene oxide increased with H_2_SO_4_ concentration as observed by Fourier-transformed infrared spectroscopy and X-ray photoelectron spectroscopy. It was observed that GO lost primarily epoxide functional groups from 40.4 to 9.7% and obtaining 69.8% carbon when using 12 M H_2_SO_4_, without leaving sulfur doping. Additionally, the appearance of hexagonal domain structures observed in transmission electron microscopy and analyzed by selected area electron diffraction patterns confirmed the improvement in graphitization. Although the addition of SO_3_ in H_2_SO_4_ improved the GO reduction with 74% carbon, as measured by XPS, the use of SO_3_ introduced sulfur doping of 1.3%. RGO produced with sulfuric acid was compared with a sample obtained via ultraviolet (UV) irradiation, a very common reduction route, by observing that the RGO produced with sulfuric acid had a higher C/O ratio than the material reduced by UV irradiation. This work showed that sulfuric acid can be used as a single-step reducing agent for RGO without sulfur contamination.

## 1. Introduction

Graphene is a 2D C nanomaterial with *sp^2^* hybridization that has a wide range of industrial applications, including the production of ion-selective graphene oxide (GO) membranes for desalination and salinity gradient energy, radioactive wastewater treatment, the production of heavy water, and others [1,2,3,4]. Generally, this material is used as graphene nanoplatelets (GNPs) due to the possibility of producing it directly from graphite in larger volumes at a lower cost. For example, it can be produced from mechanical exfoliation by using attrition milling [5,6,7] or from the oxidation, exfoliation (GO), and reduction (reduced GO [RGO]) of graphite [8,9,10,11,12]. Although GO reduction is a suitable route for the mass production of graphene, the restoration of the graphene lattice by eliminating the O functional groups with chemical treatments or ultraviolet (UV) light results in GNPs with structural defects and different concentrations of O, depending on the processing route. For example, hydrazine leaves a N doping that is difficult to remove from the final product [9,10], whereas the reduction that uses UV light can cause graphene sheet fragmentation [13]. Therefore, it is desirable to find simple and cost-effective routes to produce RGO.

Sulfur-containing compounds were also previously suggested as reducing agents for RGO production. S-containing compounds such as sodium hydrogen sulfite (NaHSO_3_), sodium sulfide nonahydrate (Na_2_S·9H_2_O), sodium thiosulfate (Na_2_S_2_O_3_), and thionyl chloride (SOCl_2_), have been used for the reduction of GO. These compounds were able to decrease the concentration of hydroxyl and epoxide groups since the mass percentage of oxygen decreased from 50.5 to 15.3% with NaHSO_3_, 18.6% with Na_2_S, 25% with Na_2_S_2_O_3_, and 13.4% with SOCl_2_ [10]. However, their use also resulted in the presence of sulfur with values from 0.79 to 1.98 mass% [10]. Other methods of chemical reduction of GO involved two-step processes such as diluted and concentrated H_2_SO_4_, as well as H_2_SO_4_ and sodium borohydride (NaBH_4_) [8]. In these studies, the proportion of epoxide and hydroxyl groups decreased from 36 to 12% with concentrated H_2_SO_4_ and until 9% with H_2_SO_4_ and NaBH_4_ [8]. Dilute sulfuric acid has been used to open the epoxy ring and transform it into hydroxyl groups, to later remove them with concentrated sulfuric acid at 120 °C [8]. In order to find a more eco-friendly method, H_2_SO_4_ has been combined with organic solvents such as dimethyl sulfoxide (DMSO) and N, N’-dimethylformamide (DMF), also in a two-step process. It was possible to increase the carbon percentage from 50.6% to 80.8 and 84.5% with DMF and DMSO, respectively. The C/O ratio was also increased from 1.4% to 6.7% for DMF and 8.0% for DMSO. However, as in most chemical reduction methods, DMF resulted in nitrogen contamination of 1.5% and DMSO with 0.6% of sulfur [14].

Therefore, to simplify the reduction process, the authors studied the effects of H_2_SO_4_ concentration and H_2_SO_4_ with 30% of sulfur trioxide (SO_3_), commonly known as fuming sulfuric acid, for the reduction of GO in a single-step process. Reduction was observed to strongly depend on H_2_SO_4_ concentration since, at 12 M, the graphene structure was reestablished due to the loss of functional groups, such as epoxide, showing that 12 M H_2_SO_4_ can be used to reduce GO without the need for a preliminary step using diluted H_2_SO_4_ as previously reported. However, the results show that adding SO_3_ into H_2_SO_4_ not only slightly improved the reduction of graphene oxide, it also resulted in the doping of the final product with 1.3 at% sulfur.

## 2. Experimental Section

### 2.1. Synthesis of GO and RGO

GO was prepared by using the improved Hummers method proposed by Marcano et al. [15]. A mixture of sulfuric acid (95–98%, Sigma-Aldrich, St. Louis, MO, USA) and phosphoric acid (85.8%, J. T. Beaker Phillipsburg, NJ, USA) with a 9:1 volume ratio was prepared with the addition of 3 g of graphite flakes (95%, Sigma-Aldrich St. Louis, MO, USA). This mixture was stirred for 30 min before adding 6 g of potassium permanganate (99%, Sigma-Aldrich St. Louis, MO, USA) at a temperature of 50 °C. After 24 h, the temperature of the mixture dropped to 2 °C, and 3 mL of hydrogen peroxide (30%, Sigma-Aldrich St. Louis, MO, USA) were carefully added. Subsequently, deionized water was added until a pH of 1 was reached. The solid material was washed twice with a solution of HCl at 30% *v*/*v* (36.5–38% Sigma-Aldrich St. Louis, MO, USA), deionized water, and finally ethanol (99.5%, Analytyka Mexico State, Mexico). The sample was coagulated by using ethyl ether (99%, Sigma-Aldrich St. Louis, MO, USA) and centrifuged for 30 min at 3500 RPM (XC-2450 PREMIERE). The solid was then dispersed in ethanol (99.5%, Analytyka Mexico State, Mexico) with an ultrasonic bath (Branson 3800 40 kHz/230–240 V Danbury, CT, USA) for 1 h to exfoliate the material. The final GO was then dried at 80 °C for 12 h and milled before its characterization [16]. The pH of a suspension of GO in deionized water generally is between 2 and 3; measured using an OHAUS potentiometer, model starter 2100 (Parsippany, NJ, USA).

RGO was obtained by mixing 0.3 g of GO in a balloon flask with 50 mL of H_2_SO_4_ (≥97.5%, Sigma-Aldrich St. Louis, MO, USA) at 1.5, 5, and 12 M, in addition to a 12 M H_2_SO_4_ with 30% of sulfur trioxide (SO_3_)(as received fuming sulfuric acid, Sigma-Aldrich, 28–32% St. Louis, MO, USA), and kept under reflux at 90 °C for 24 h. Afterward, RGO was separated by using a centrifuge (PREMIERE XC-2450) at 3500 RPM for 15 min. The solid was washed several times with deionized water until a pH of 6 was reached. Finally, the solid was dried at 80 °C for 12 h and milled in an agate mortar. For comparison purposes, RGO was also prepared by using a UV reduction process in water with three UV lamps of 7.2 W (Tecno Lite, F8T5BLB) with a wavelength of 368 nm. The highest irradiance inside the box was measured at 74 μW/cm^2^ with a photodiode OPHIR, DP-300 series (Jerusalem, Israel). The GO suspensions were prepared in borosilicate glass flasks by using 12.5 mg of GO in 10 mL of deionized water. The pH was adjusted to 7 with HCl or NaOH solutions, as necessary. The suspensions were then placed into the black box container with constant stirring for 120 h. To avoid water evaporation, the temperature was kept at 25 °C by using a recirculating chiller (IKA RC 5 basic Staufen, Germany) [13].

### 2.2. Characterization

The functional groups and C content of GO and RGO were characterized by Fourier-transform infrared spectroscopy (FTIR) (PerkinElmer Frontier FTIR/NIR, Waltham, MA, USA) and by X-ray photoelectron spectroscopy (XPS) (PHI VersaProbe II, Chanhassen, MN, USA) with a 2 × 10^−8^ mTorr vacuum chamber, an Al anode as an X-ray monochromatic source with radiation energy of 1486.6 eV, and an analysis range from 1400 to 0 eV. The processing of the XPS data of the C1s region was carried out in the CasaXPS program using a Shirley background [17]. Carbon with *sp*^2^ and *sp*^3^ hybridization (C-C/C=C) was assigned to the binding energy of 284.9 eV and adjusted with a Gaussian function with 30% of a Lorentzian function with symmetric shape. The signals corresponding to the oxygenated functional groups were assigned to 285.9 eV (C-OH), 286.9 eV (C-O-C), 288.2 eV (C=O), and 289.3 eV (COOH) [12,18,19,20,21] which were also fitted with a Gaussian with 30% Lorentzian function and with symmetric shape, whereas the binding energy was calibrated using the carbon energy at 284 eV [17]. The microstructure and elemental analyses were also characterized by an Aberration corrected FEI TITAN (Hillsboro, OR, USA) transmission electron microscope (TEM) operated at 300 kV. Raman spectroscopy was also performed by using a RENISHAW (Wotton-under-Edge, Gloucestershire, UK) inVia Microscope with a laser excitation wavelength of 514 nm and a 50× lens. The Fityk program and a mixture of Gaussian (G, D2, and D3 bands) and Lorentzian (D, D4, and C bands) functions were used to fit the Raman bands [22].

## 3. Results and Discussion

Figure 1 shows the FTIR spectra of the as-produced GO and the RGO treated with H_2_SO_4_ and fuming H_2_SO_4_. The GO spectrum (Figure 1a) showed a broad and intense band close to 3500 cm^−1^ that corresponds to the stretching mode of the hydroxyl (-OH) functional group, whereas the signal at 1725 cm^−1^ was associated with the carbonyl group (C=O). Similarly, the signal at 1600 cm^−1^ was assigned to the presence of carbon with *sp*^2^ hybridization, whereas the band at 1500 cm^−1^ corresponds to carbon with *sp*^3^ hybridization, resulting from the double bond breaking of graphite. Finally, two bands around 1040 and 1200 cm^−1^ were observed corresponding to C-O bonds assigned to the hydroxyl (C-OH) and epoxide (C-O-C) groups, respectively [23].

Once GO was treated with H_2_SO_4_, the intensity of all the bands gradually decreased with H_2_SO_4_ concentration. The most evident decrease in intensity was observed after the chemical treatment with 12 M H_2_SO_4_ and fuming H_2_SO_4_ (Figure 1d,e); at this concentration, for example, the band at 3500 cm^−1^ disappeared. This behavior is associated with the loss of the GO functional groups since pure C is not infrared active [24]. Conversely, although the RGO obtained by UV irradiation also showed a reduction in band intensity, after 120 h under UV irradiation, it was still possible to identify the presence of C-OH, C=O, C-O-C, and COOH functional groups, obtaining only similar results as RGO treated with 1.5 M H_2_SO_4_.

Figure 2 shows the XPS spectra of all the RGO produced. In the deconvoluted spectrum of as-produced GO (Figure 2a), it was possible to identify the bands at 284.9 ± 0.2 eV which correspond to the C structure with *sp*^2^ and *sp*^3^ hybridization [12,19,20,21]. Furthermore, the signals corresponding to C-OH (286.0 ± 0.1 eV), C-O-C (287.1 ± 0.2 eV), C=O (288.2 ± 0.05 eV), and COOH (289.2 ± 0.1 eV) were also identified [18]. Similar to FTIR, it was observed that the intensity of the bands that correspond to the oxygenated functional groups decreased with H_2_SO_4_ concentration (Figure 2b–e). The C-O-C group reduced its concentration gradually from 40.4% in as-produced GO to 9.7% in RGO with 12 M H_2_SO_4_ (Table 1). On the other hand, the C-OH group increased its percentage from 4.3 to 5.5% after been treated with 1.5 M H_2_SO_4_ but decreased to 1.9% with a 5 M treatment. Similar behavior was observed for the C=O and COOH functional groups, which increased from 9.6 and 2.3% to 16.4 and 3.7%, respectively, with 1.5 H_2_SO_4_, and decreased to 9.0 and 3.6% with 5 M H_2_SO_4_. Nevertheless, C=O decreased again to 1.8% and COOH remained at 3.6% after been treated with 12 M H_2_SO_4_. All these changes in the concentration of oxygenated functional groups were reflected in the GO carbon structure since the as-produced GO had 43.3% of C-C/C=C due to its high oxidation degree due to the use of a high concentration of oxidating agent (KMnO_3_) [16]; however, this value changed to 69.8% after treatment with 12 M H_2_SO_4_. Previously, Kim et al. [8] reported GO reduction by using H_2_SO_4_ and NaBH_4_ as an initial step. An increase of C=C/C-C bonds from 43 to 69% was reported from the loss of oxygenated functional groups that were similar values to this work, showing that the single use of H_2_SO_4_ can increase C content from 56.1 to 79.9 at% (Table 1). Furthermore, some authors have reported that the NaBH_4_ compound might destroy the graphene-like structure that, in addition to N doping, generates unfavorable effects on graphene conduction and flexibility [12]. In this case, even when fuming H_2_SO_4_ increased the C-C/C=C content to 74% and decreased the functional groups C-O-C, C=O, and COOH to 2.9, 7.5, and 2.2%, respectively, the presence of S was observed in 1.3 at%. Conversely, none of the samples produced with only H_2_SO_4_ showed the presence of sulfur contamination (Appendix A).

On the other hand, the RGO obtained by UV irradiation (Figure 2f) also showed an increase in the *sp*^2^ and *sp*^3^ hybridization band from 43.3 to 61.4% and a decrease in the bands that correspond to oxygenated functional groups, primarily the band that corresponds to C-O-C but not at the same level as GO treated with 12 M H_2_SO_4_. The C-O-C decreased its concentration from 40.4 to 22.9% in as-produced GO and irradiated RGO, respectively. Additionally, C-OH increased its concentration from 4.3 to 6% in as-produced and irradiated GO, respectively. Furthermore, C=O and COOH groups maintain their concentration in values around 8.6 and 1.1% These variations in composition were also reflected on the C/O ratios measured for each sample (Table 1) since the value increased from 1.4 to 4 for H_2_SO_4_ 12 M after losing ~50% of the original O present in GO, whereas the RGO obtained with UV light only reached a value of 2.2 C/O with a loss of ~30% O.

The microstructure of GO and RGO is shown in Figure 3. Figure 3a shows the as-produced GO layer that is ~20 μm long and has the appearance of wrinkles generated due to the tension in the network created by the presence of oxygenated functional groups [25]. The disorder in the GO C structure was evident from the formation of concentric rings in its selected area electron diffraction pattern (SAEDP) [26]. This microstructure changed after treatment with 12 M H_2_SO_4_, as observed in Figure 3b,c. Figure 3b shows the bright field of the hexagonal crystalline structure taken in a zone axis orientation, as shown in the SAEDP attached. The main importance of this image is remarked on in Figure 3c in which a dark field image of the same area shows the differentiated hexagonal domain structures obtained with the conical dark field technique, which allows images to be obtained from the entire set of intensities in the diffraction pattern. The RGO appeared to be formed of ~5 µm particles. Dark field images showed the formation of areas with more crystalline structure since the bright areas in the dark field correspond to the settlement of the C domains in the same direction. Furthermore, the SAED pattern shows a hexagonal diffraction pattern that confirms the restructuring of graphene and the formation of areas with good crystallinity [27].

The Raman spectra of GO and RGO are shown in Figure 4 and Appendix A. The Raman spectrum of as-produced GO in Figure 4a shows the presence of the G band in 1591.93 cm^−1^ generated by the stretching of *sp*^2^ bonds from an ideal graphite lattice associated with the first-order Raman mode E_2g_. It was also observed that the D band in 1356.04 cm^−1^, which is attributed to a disordered graphitic lattice due to the conversion of *sp*^2^ to *sp*^3^, bonds by the graphene oxidation, which is associated with the vibrational mode A_1g_ [28,29] (Appendix A). Three more bands were also identified as D2 at ~1620 cm^−1^, D3 at ~1500 cm^−1^, and D4 at ~1200 cm^−1^ (Appendix A) [28], which are related to the disorder in the basal plane and the presence of amorphous C [28,30,31]. Additionally, the C band close to 1700 cm^−1^, which is generally attributed to the presence of C=O functional groups, was also identified [22]. The G and D bands were also observed in the RGO spectra treated with 12 M H_2_SO_4_; however, their position shifted to 1599.28 cm^−1^ for the G band and 1357.97 cm^−1^ for the D band (Appendix A). Additionally, for the UV irradiated RGO, the G and D bands shifted to 1593.78 and 1357.03 cm^−1^, respectively. The shift of the G band towards the position found in graphite of approximately ~1581 cm^−1^ [9] is generally indicative of the restoration of the *sp*^2^ structure in carbon [32]. However, our results show that the G bands in RGO sifted towards higher wavenumbers. Schüpfer et al. observed similar behavior when studying the transition from *sp*^3^ to *sp*^2^ hybridization in heat-treated graphitic and non-graphitic carbon, with a G band close to 1600 cm^−1^ [33].

The ratio between the intensity of the D and G bands (I_D_/I_G_) was also calculated since it is generally used to measure structural disorder. The I_D_/I_G_ ratio decreased from 1.62 to 1.57 in RGO treated with 12 M H_2_SO_4_, which agrees with previous reports on RGO [8,9,10]. It was also reported that GO reduction leads to the formation of numerous *sp*^2^ domains, which are smaller than the average-sized domains in graphene [8,9,10]. This is consistent with the formation of nanocrystalline graphite in which not only does the D band appear but the G band shifts toward approximately 1600 cm^−1^, and the I_D_/I_G_ ratio increase to almost 2 [34]. Conversely, for UV irradiated RGO, the ratio I_D_/I_G_ increased to 1.83. This increase could suggest the generation of graphitic domains and the breakage of the graphene sheets, as observed in TEM. Similarly, for RGO treated with fuming H_2_SO_4_ 12 M, the G band was located at 1599.28 cm^−1^ and the D band at 1356.04 cm^−1^, whereas the I_D_/I_G_ ratio decreased to 1.45.

The changes in graphitization were also evident in the variations of the full width at half maximum (FWHM) from the G band, a measure often used as a parameter of graphitization [35]. The FWHM values of the samples produced are shown in Appendix A. The FWHM of as-produced GO and RGO with 12 M H_2_SO_4_ decreased from 57.92 cm^−1^ to 54.12 cm^−1^, supporting the increase in graphitization due to the reduction induced by H_2_SO_4_. Furthermore, the band D2 decreases its intensity, suggesting the disappearance of 5-8-5 clusters, which are associated with this band [31].

The proposed reaction mechanism for the dehydroxylation in RGO by using H_2_SO_4_ is shown in Appendix A. It is suggested that the -OH is protonated by an H^+^ from H_2_SO_4_ [36], which occurs due to the electrostatic interaction between the positive charge of hydrogen in H_2_SO_4_ and the negative charge of oxygen in the -OH functional group [11]. Once the acid yields one of its protons to the hydroxyl, a water molecule is formed in addition to HSO_4_^–^. Subsequently, the water molecule is also released, leaving a carbon with a positive charge that forms a double bond with the neighboring carbon. This reaction was proposed to be favorable and has an activation energy of 7.7 kcal/mol [11]. On the other hand, it was proposed that the de-epoxidation occurs by opening the epoxide ring [11,12]. Initially, oxygen is protonated, which causes the opening of the ring, resulting in the formation of a C-OH functional and a water molecule. Finally, this water molecule releases an H^+^, producing another -OH group (Appendix A) [11].

As previously reported, during GO oxidation, the first functional group to be generated is -OH, and its subsequent oxidation results in the C-O-C, C=O, and COOH groups [23]. The reduction process occurs similarly since, in addition to the loss of -OH groups, there is the transformation of the C-O-C group into the -OH, which was previously described as de-epoxidation [11,12]. This explains what was observed by XPS (Table 1) in which the percentage of the -OH and C-O-C group decreased after the treatment with a low concentration of H_2_SO_4_. However, as the acid concentration increases, the percentage of the -OH group increases, and the percentage of C-O-C decrease significantly. On the other hand, it was also reported that the reduction of the C=O and COOH groups by S-containing compounds requires high activation energy, making it easier to reduce the functional groups inside the basal plane [11].

## 4. Conclusions

The chemical reduction of GO by using H_2_SO_4_ was achieved in different degrees, depending on its concentration. The optimal concentration for the production of RGO was 12 M since the carbon structure was reestablished through the loss of the oxygenated functional groups such as an epoxide. A reaction mechanism was proposed to illustrate the transformation of the epoxide group into the hydroxyl (-OH) group and its subsequent elimination by a dehydration process due to the action of sulfuric acid. Furthermore, no sulfur doping was observed, a common problem in chemical reduction methods for GO with sulfur-containing compounds such as sodium hydrogen sulfite (NaHSO_3_) or with nitrogen-containing compounds such as hydrazine (N_2_H_4_). The use of fuming sulfuric acid also showed some improvement on the reduction of GO by reaching a carbon content of 74%, however, XPS suggested that it also led to the contamination of RGO with 1.3% sulfur. Using H_2_SO_4_ is a promising method for reducing GO to obtain RGO since it can be applied as a single-step.

## Figures and Tables

**Figure 1 materials-14-00059-f001:**
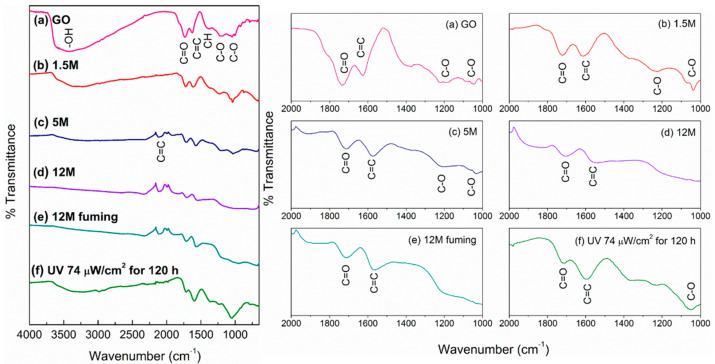
Infrared spectra (**left**) and details of the IR spectra from 2000 to 1000 cm^−1^ (**right**) of (**a**) as-produced GO and RGO with H_2_SO_4_ at (**b**) 1.5 M, (**c**) 5 M, (**d**) 12 M, (**e**) fuming H_2_SO_4_ 12 M, and (**f**) RGO irradiated at 74 μW/cm^2^ for 120 h.

**Figure 2 materials-14-00059-f002:**
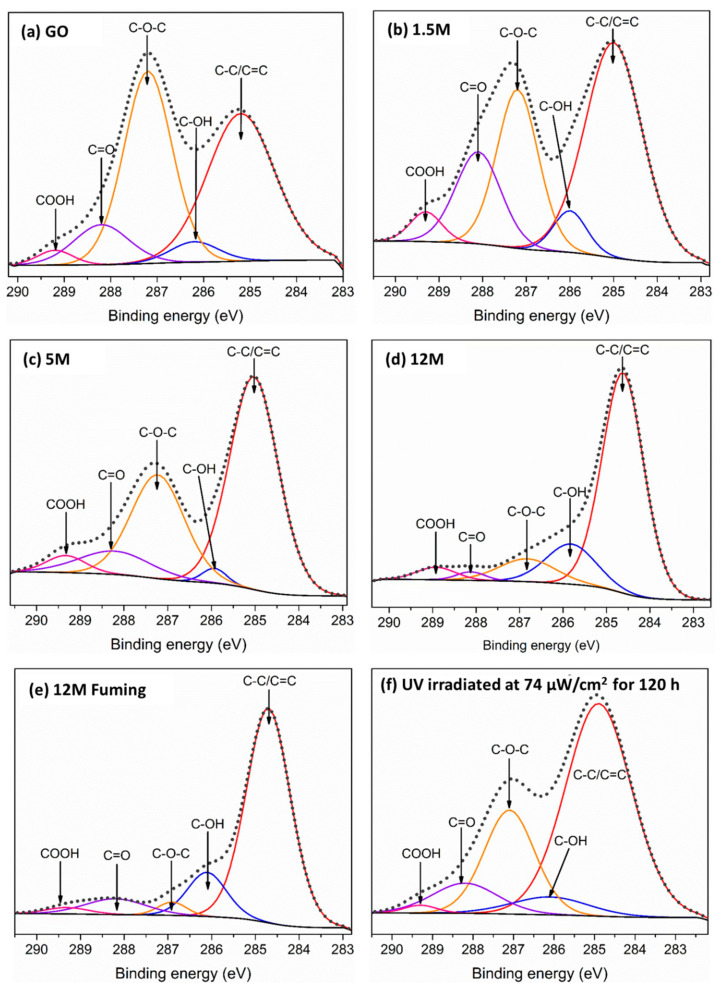
XPS spectra of (**a**) as-produced GO and RGO with H_2_SO_4_ in (**b**) 1.5 M, (**c**) 5 M, and (**d**) 12 M concentrations; (**e**) RGO with fuming H_2_SO_4_ in 12 M concentration; and (**f**) RGO irradiated at 74 μW/cm^2^ for 120 h.

**Figure 3 materials-14-00059-f003:**
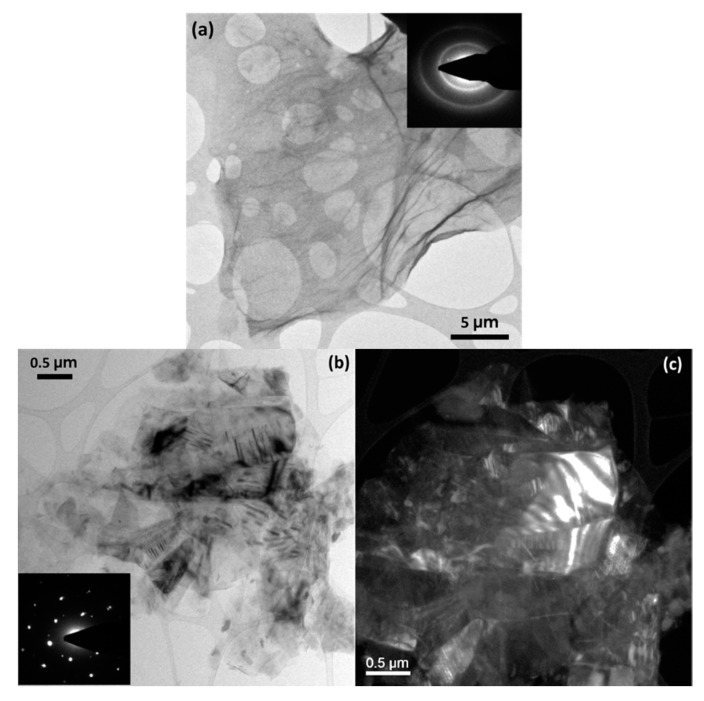
TEM micrographs of (**a**) as-produced GO with SAED pattern and (**b**,**c**) bright and dark field of RGO H_2_SO_4_ 12 M with SAED pattern.

**Figure 4 materials-14-00059-f004:**
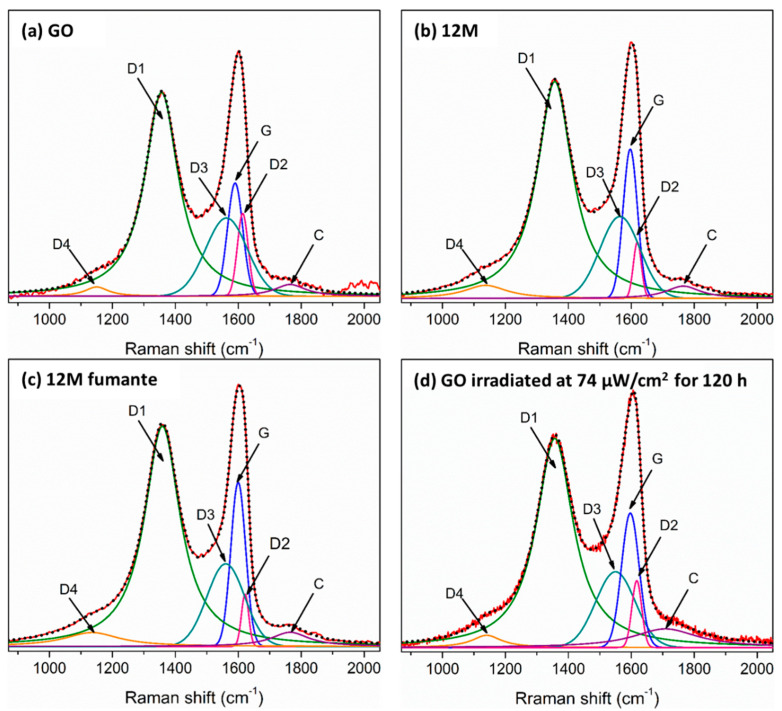
Raman spectroscopy spectra of (**a**) as-produced GO, (**b**) RGO treated with H_2_SO_4_ in 12 M concentration, (**c**) RGO treated with fuming H_2_SO_4_ in 12 M concentration, and (**d**) RGO irradiated at 74 μW/cm^2^ for 120 h.

**Table 1 materials-14-00059-t001:** Quantification of functional groups in as-produced GO and RGO with H_2_SO_4_ and fuming H_2_SO_4_ by XPS.

Bond	GO	1.5 M	5 M	12 M	Fuming 12 M	74 μW/cm^2^ 120 h
	%	%	%	%	%	%
C-C/C=C	43.3	48.6	55.8	69.8	74	61.4
C-OH	4.3	5.5	1.9	15.1	13.4	6
C-O-C	40.4	25.2	29.7	9.7	2.9	22.9
C=O	9.6	16.4	9.0	1.8	7.5	8.6
COOH	2.3	3.7	3.6	3.6	2.2	1.1
Carbon At. %	56.1	70.1	71.9	79.9	77.8	64.9
Oxygen At. %	41.3	29.9	28.1	20.1	20.9	29.8
C/O	1.4	2.3	2.6	4.0	3.7	2.2

## Data Availability

Data is contained within the article or Appendix A.

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
