# Peer review of "Controlled Reduction of Graphene Oxide Using Sulfuric Acid"

_materials, 2020, doi:10.3390/ma14010059_

Round 1

Reviewer 1 Report

The authors used sulfuric acid as the reducing agent for the production of RGO. The reduction can be improved with 12 M H2SO4, but has no effect with SO3. The reaction mechanisms regarding H2SO4, SO3, and UV radiation are discussed and meaningful. The characterizations by Raman, XPS, and TEM are helpful to understand the mechanism. The result is significant and well discussed. However, some points need to be clarified before the manuscript could be considered for publication.

1. I would like to know how the FWHMs of each deconvolution peak in Figure 2 are determined? For example, sp3 peaks in different subfigures have different FWHMs. In Raman, the FWHM of the G band means degrees of disorder. What does it mean in XPS ? Some sp2 peak positions are less than 284.0 eV, whereas some are larger than 284.0 eV. How is the sp2 peak position determined and what is the scientific rationale behind it?

2. Line 157: The authors stated "The G and D bands were also observed in the RGO spectra treated with 12 M H2SO4; however, their position shifted to 1,599.28 cm-1 for the G band and 1,357.97 cm-1 for the D band (Table S2). Additionally, for the UV irradiated RGO, the G and D bands shifted to 1,593.78 and 1,357.03 cm-1, respectively. The shift of the G band toward higher wavenumber values suggests the restoration of the in-plane sp2 domains since it approaches the position that it has in graphite (~1,581 cm-1)." The shift of the G band toward higher wavenumber values than which? It seems that the G band for UV irradiated RGO (1591.93 -> 1593.78 cm-1) has lower wavenumber than that for 12 M H2SO4 treated RGO (1591.93 -> 1599.28 cm-1). Besides, in the text, "the shift of the G band toward higher wavenumber values" seems to be away from graphite (~1,581 cm-1). Why do the authors stated "approach the position that it has in graphite (~1,581 cm-1)." The authors should clarify it.

3. Line 227: In conclusion, the authors stated "Using H2SO4 is a promissory method for reducing GO to obtain RGO oxide since it can be applied as a single step without any evidence of S doping." The word "oxide" can be removed from the sentence. By the way, how do the authors make the conclusion that no S doping is in RGO. I don't see any evidence in the text. Maybe a narrow scan of the S 1s core level of the RGO by XPS can be an evidence.

4. Line 116: "...and 8.0 with 5 M H2SO4,.." It seems that "%" is missing after 8.0.

Author Response

Dear Editor,

We wish to submit the revised version of our manuscript materials-997842 entitled "Controlled Reduction of Graphene Oxide Using Sulfuric Acid", for consideration to the journal of Materials. The corrections to the manuscript and response to the reviewers are listed below (all changes are highlighted in yellow in the manuscript):

REVIEWER #1

The authors used sulfuric acid as the reducing agent for the production of RGO. The reduction can be improved with 12 M H2SO4 but does not effect with SO3. The reaction mechanisms regarding H2SO4, SO3, and UV radiation are discussed and meaningful. The characterizations by Raman, XPS, and TEM are helpful to understand the mechanism. The result is significant and well discussed. However, some points need to be clarified before the manuscript could be considered for publication.

  1. I would like to know how the FWHMs of each deconvolution peak in Figure 2 are determined? For example, sp3 peaks in different subfigures have different FWHMs. In Raman, the FWHM of the G band means degrees of disorder. What does it mean in XPS? Some sp2 peak positions are less than 284.0 eV, whereas some are larger than 284.0 eV. How is the sp2 peak position determined and what is the scientific rationale behind it?

Response: For the analysis of the XPS data, the Casa XPS program was used in which the FHWM was adjusted to 1 eV. The change in the FWHM is attributed among other things to a change in the number of bonds in the functional group. See for example [17] Fairley (2009).

Regarding the position of the sp2 bond, it was established according to the reviewed literature which ranges from 284.4 to 284.6, see for example [19] Stankovich (2006); [120] Ganguly (2011), [21] Ren (2011) and [12] Pei (2010). The band assigned in our work is within those previously used by these and other authors. On the other hand, the decrease in bond energy is attributed to the atomic relaxation process that is related to the rearrangement of electrons (Lines 105-112 and 139-143).

  1. Line 157: The authors stated "The G and D bands were also observed in the RGO spectra treated with 12 M H2SO4; however, their position shifted to 1,599.28 cm-1 for the G band and 1,357.97 cm-1 for the D band (Table S2). Additionally, for the UV irradiated RGO, the G, and D bands shifted to 1,593.78 and 1,357.03 cm-1, respectively. The shift of the G band toward higher wavenumber values suggests the restoration of the in-plane sp2 domains since it approaches the position that it has in graphite (~1,581 cm-1)." The shift of the G band toward higher wavenumber values than which? It seems that the G band for UV irradiated RGO (1591.93 -> 1593.78 cm-1) has a lower wavenumber than that for 12 M H2SO4 treated RGO (1591.93 -> 1599.28 cm-1). Besides, in the text, "the shift of the G band toward higher wavenumber values" seems to be away from graphite (~1,581 cm-1). Why do the authors state "approach the position that it has in graphite (~1,581 cm-1)." The authors should clarify it.

Response: The text was modified to clarify the displacement of the bands, lines 200-204. The displacement of the G band towards wavenumbers greater than the position in which it is found in the graphite (1581 cm-1), it is considered that the material loses sp2 hybridization, however, in this case, the XPS and TEM analysis shows that sp2 hybridized domains are being restored. Was reported by [34] Schüpfer (2020) that areas of carbon with a mixture of sp2 and sp3 hybridization, which are also very close to sp2 hybridized domains results in the G band close to 1600 cm-1.

  1. Line 227: In conclusion, the authors stated "Using H2SO4 is a promissory method for reducing GO to obtain RGO oxide since it can be applied as a single step without any evidence of S doping." The word "oxide" can be removed from the sentence. By the way, how do the authors conclude that no S doping is in RGO?. I don't see any evidence in the text. Maybe a narrow scan of the S 1s core level of the RGO by XPS can be evidence.

Response: The word "oxide" has been removed from the conclusion, and it was decided to add a figure, in the supplementary material (Figure S1), of the complete XPS scan in order to show all the signals observed in the RGO spectra, this is mentioned in the main text in lines 160-163.

  1. Line 116: "...and 8.0 with 5 M H2SO4..." It seems that "%" is missing after 8.0.

Response: The error in that statement was corrected, line 147. Besides, some values were changed due to the modification in XPS analysis.

REVIEWER #2:

  1. Lines 42-44: In the introduction section, more information from the previously published manuscripts must be mentioned and discussed to highlight the novelties in this study.  For example,  in reference 8 „C A R B ON 5 0 ( 2 0 1 2 ) 3 2 2 9 –3 2 3 2”,  the authors have mentioned: In the method described here, GO was reduced by a two-step process solely with sulfuric acid to produce rGO with a reduction level comparable to that obtainable by previously reported methods.” In this study [8], authors have also used just H2SO4 and in different concentrations in two steps as diluted and concentrated acids.

Response: The introduction has been extended referencing other previous work using sulfur-containing compounds; lines 45 to 60. The work mentioned by the reviewer was already referenced in the original version, however, it was further described in the introduction. As we have now stated in the introduction, lines 50 to 55, our results show that the two-step process to reduce GO is unnecessary and that 12 M H2SO4 can be used to effectively reduce GO without the addition of S in its structure. Additionally, we stress that the use of SO3 as fuming sulfuric acid also results in undesirable S doping.   

  1. Line 56: Please replace the “K” with “potassium”.

Response: The symbol K was replaced by the full element name, line 76.

  1. Line 66: How is the” 12 M H2SO4 with 30% SO3” prepared?

Response: Fuming sulfuric acid (sulfuric acid with 30% SO3) was purchased from Sigma Aldrich and used as received. This information was included in lines 87-89 in the Experimental section.

  1. In the abstract is mentioned that: “Conversely, adding SO3 did not improve the GO reduction since the material reoxidized as the hydroxyl, carbonyl, and carboxylic acid groups increased from 18.0, 3.0, and 0.7% to 36.9, 6.0, and 7.8%, respectively”. However, Figure 1d and e are the same. It means FTIR could not help to see the differences. These percentages were calculated from XPS results. Using obtained percentages from deconvoluted graphs is a very tricky way to conclude a general phenomenon. Please make it clearer e.g. mention that in the abstract.

Response: The abstract was modified to mention that FTIR and XPS were used to study the effect of sulfuric acid concentration. Additionally, it should be stated that FTIR is a semiquantitative method therefore, their results although similar cannot be as quantifiable as XPS, which is a technique that is commonly used for the quantification of functional groups and characterization of RGO. Additionally, in Figure 2, the zoom to the region of the functional groups was added, in which the decrease in intensity of these bands is observed more clearly.

  1. For XPS analysis, is the binding energy calibrated? How? Please mention it. How were the fits of the high-resolution spectra performed? Please provide the information.

Response: For the binding energy calibration was used the energy of carbon at 284 eV and for the analysis of the XPS data, the CasaXPS program was used with a Shirley-type background. The bands were fitted at 284.9 eV (C=C/(C-C), 285.9 eV (C-OH), 286.9 eV (C-O-C), 288.2 eV (C=O), and 289.3 eV (COOH), reference [18] Perrozzi (2014). A more detailed description in this regard was added in the article on lines 105 to 112.

  1. In XPS results, please clarify, why the sp3 bond is bigger than sp2?

Response: Following other reviewers’ comments, the carbon signal was described as a single signal for C=C/C-C. The discussion was modified to reflect this change. Nevertheless, the high concentration of sp3 carbon in GO originates from its high oxidation degree. For this particular GO, we used a higher concentration of oxidation agent (KMnO3) than commonly used, in order to obtain a highly oxidized sample, see reference [24] Reynosa-Martínez (2020). We have added a comment in lines 153-153 and a reference to address this comment.

  1. Line 119-120: Please be careful that Kim et al.[8] reported GO reduction by using H2SO4 and NaBH4 as an initial step, to compare their method (comment 1) with the previously reported method.

Response: The work of Kim et al. was already mentioned in the introduction (lines 50-55) but their work was further discussed in the revised version, lines 45-60.   

  1. In table1, How is C/O calculated?

Response: The carbon-oxygen ratio (C/O) was calculated from the atomic percentages obtained from the complete XPS scan. The atomic percentages of carbon and oxygen have been added in Table 1.

  1. Line 195-200 and Figure S5: 1) in Figure S5, please show the mechanism on a graphene structure. 2) What does it mean: “It is suggested that the -OH is protonated by an H+ from H2SO4, which occurs due to the electrostatic interaction between the negative charge of oxygen in H2SO4 and the positive charge of H in the -OH functional group.” From the basic chemistry, protonation of -OH occurs due to the negative charge of oxygen in –OH and positively charged H produced by acid. Isn’t it? 3) Please replace “Paso 3” with “step 3”. Furthermore, step 3 is not discussed in the text. This mechanism is the authors’ hypothesis or it is reported in other references. Can vinyl groups be produced by adding just sulphuric acid? Please discuss it.  

Response: The structure of graphene oxide was added to the proposed mechanism (now Figure S6), in order to better represent the effect of sulfuric acid on its functional groups. The complete discussion of the reaction mechanism, lines 237 to 243, was also added and the error related to the charges of the hydroxyl group and the acid proton, in lines 238 to 240, was corrected.

It has been reported that this acid is capable of dehydrating graphene oxide, through the protonation of the hydroxyl group and the subsequent release of the water molecule, see for example lines 53-55, reference [8] Kim (2012) in the manuscript. In this work, it is proposed that, after the release of the water molecule, a carbocation is formed that stabilizes by forming a double bond with the carbon neighbor.

  1. Figure S6: please show the mechanism on a graphene structure.

Response: The graphene structure was added to all proposed mechanisms; Figures S6 and S7.

  1. Figure S5 and Figure S7: What is the main role of H2SO4: -OH -> = or –OH->=O? Please discuss it.

Response: The effect of sulfuric acid does not occur selectively as it might affect different functional groups as described in the manuscript in lines 248 to 251. At the basal plane, the groups that can be transformed are hydroxyl and epoxides, while at the edges it is hydroxyl since the carboxyl and carbonyls are more stable, reference [11] Su (2014).

  1. In reference 10 “J. Phys. Chem. C 2010, 114, 19885–19890”, an interesting mechanism for the possible reduction of GO by NaHSO3 is proposed. In this mechanism, SO3 is produced as one of the final products. What is the possible reduction mechanism with H2SO4? Could SO3 be produced as one final product? When yes, that can explain, why adding SO3 has a negative effect on your process.

Response: The SO3 group is not considered as a final product or by-product of the reaction, since sulfuric acid in an aqueous medium is capable of recovering the released proton in a kind of acid-base equilibrium therefore it would not become SO3, see for example [37] Shin (2019). Nevertheless, the reaction mechanism proposed has been removed from the main text and the supporting information to avoid confusion on the effect of acid on graphene oxide because it is no longer necessary in the revised version and following other reviewers’ comments.

Reviewer 2 Report

-Lines 42-44: In the introduction section, more information from the previously published manuscripts must be mentioned and discussed to highlight the novelties in this study.  For example,  in reference 8 „C A R B ON 5 0 ( 2 0 1 2 ) 3 2 2 9 –3 2 3 2”,  the authors has mentioned: In the method described here, GO was reduced by a two step process solely with sulfuric acid to produce rGO with a reduction level comparable to that obtainable by previously reported methods.” In this study [8], authors have also used just H2SO4 and in different concentrations in two steps as diluted and concentrated acids….

-Line 56: Please replace the “K” with “potassium”.

-Line 66: How is the” 12 M H2SO4 with 30% SO3” prepared?

- In the abstract is mentioned that: “Conversely, adding SO3 did not improve the GO reduction since the material reoxidized as the hydroxyl, carbonyl, and carboxylic acid groups increased from 18.0, 3.0, and 0.7% to 36.9, 6.0, and 7.8%, respectively”. However, Figure 1d and e are the same. It means, FTIR could not help to see the differences. These percentages were calculated from XPS results. Using obtained percentages from deconvoluted graphs is a very tricky way to conclude a general phenomenon. Please make it more clear e.g. mention that in abstract….

-For XPS analysis, are the binding energy calibrated? How? Please mention it. How were the fits of the high resolution spectra performed? Please provide information.

-In XPS results, please clarify, why sp3 bond is bigger than sp2?

-Line 119-120: Please be careful that Kim et al.[8] reported GO reduction by using H2SO4 and NaBH4 as an initial step, as to compare their method (comment 1) with previously reported method.

-In table1, How is C/O calculated?

-Line 195-200 and  Figure S5: 1) in the Figure S5, please show the mechanism on a graphene structure. 2) What does it mean: “It is suggested that the -OH is protonated by an H+ from H2SO4, which occurs due to the electrostatic interaction between the negative charge of oxygen in H2SO4 and the positive charge of H in the -OH functional group.” From the basic chemistry, protonation of -OH occurs due to the negative charge of oxygen in –OH and positive charged H produced by acid. Isn’t it? 3) Please replace “paso 3” with “step 3”. Furthermore, step 3 is not discussed in the text. This mechanism is authors’ hypothesis or it is reported in other references. Can vinyl groups be produced by adding just sulphuric acid? Please discuss it.   

-Figure S6: please show the mechanism on a graphene structure.

-Figure S5 and Figure S7: What is the main role of H2SO4: -OH -> = or –OH->=O? Please discuss it.

-In refrence 10 “J. Phys. Chem. C 2010, 114, 19885–19890”, an interesting mechanism for the possible reduction of GO by NaHSO3 is proposed. In this mechanism, SO3 is produced as one of the final products. What is the possible reduction mechanism with H2SO4? Could SO3 be produced as one final product? When yes, that can explain, why adding SO3 has a negative effect in your process.

-

Author Response

Dear Editor,

We wish to submit the revised version of our manuscript materials-997842 entitled "Controlled Reduction of Graphene Oxide Using Sulfuric Acid", for consideration to the journal of Materials. The corrections to the manuscript and response to the reviewers are listed below (all changes are highlighted in yellow in the manuscript):

REVIEWER #2:

  1. Lines 42-44: In the introduction section, more information from the previously published manuscripts must be mentioned and discussed to highlight the novelties in this study.  For example,  in reference 8 „C A R B ON 5 0 ( 2 0 1 2 ) 3 2 2 9 –3 2 3 2”,  the authors have mentioned: In the method described here, GO was reduced by a two-step process solely with sulfuric acid to produce rGO with a reduction level comparable to that obtainable by previously reported methods.” In this study [8], authors have also used just H2SO4 and in different concentrations in two steps as diluted and concentrated acids.

Response: The introduction has been extended referencing other previous work using sulfur-containing compounds; lines 45 to 60. The work mentioned by the reviewer was already referenced in the original version, however, it was further described in the introduction. As we have now stated in the introduction, lines 50 to 55, our results show that the two-step process to reduce GO is unnecessary and that 12 M H2SO4 can be used to effectively reduce GO without the addition of S in its structure. Additionally, we stress that the use of SO3 as fuming sulfuric acid also results in undesirable S doping.   

  1. Line 56: Please replace the “K” with “potassium”.

Response: The symbol K was replaced by the full element name, line 76.

  1. Line 66: How is the” 12 M H2SO4 with 30% SO3” prepared?

Response: Fuming sulfuric acid (sulfuric acid with 30% SO3) was purchased from Sigma Aldrich and used as received. This information was included in lines 87-89 in the Experimental section.

  1. In the abstract is mentioned that: “Conversely, adding SO3 did not improve the GO reduction since the material reoxidized as the hydroxyl, carbonyl, and carboxylic acid groups increased from 18.0, 3.0, and 0.7% to 36.9, 6.0, and 7.8%, respectively”. However, Figure 1d and e are the same. It means FTIR could not help to see the differences. These percentages were calculated from XPS results. Using obtained percentages from deconvoluted graphs is a very tricky way to conclude a general phenomenon. Please make it clearer e.g. mention that in the abstract.

Response: The abstract was modified to mention that FTIR and XPS were used to study the effect of sulfuric acid concentration. Additionally, it should be stated that FTIR is a semiquantitative method therefore, their results although similar cannot be as quantifiable as XPS, which is a technique that is commonly used for the quantification of functional groups and characterization of RGO. Additionally, in Figure 2, the zoom to the region of the functional groups was added, in which the decrease in intensity of these bands is observed more clearly.

  1. For XPS analysis, is the binding energy calibrated? How? Please mention it. How were the fits of the high-resolution spectra performed? Please provide the information.

Response: For the binding energy calibration was used the energy of carbon at 284 eV and for the analysis of the XPS data, the CasaXPS program was used with a Shirley-type background. The bands were fitted at 284.9 eV (C=C/(C-C), 285.9 eV (C-OH), 286.9 eV (C-O-C), 288.2 eV (C=O), and 289.3 eV (COOH), reference [18] Perrozzi (2014). A more detailed description in this regard was added in the article on lines 105 to 112.

  1. In XPS results, please clarify, why the sp3 bond is bigger than sp2?

Response: Following other reviewers’ comments, the carbon signal was described as a single signal for C=C/C-C. The discussion was modified to reflect this change. Nevertheless, the high concentration of sp3 carbon in GO originates from its high oxidation degree. For this particular GO, we used a higher concentration of oxidation agent (KMnO3) than commonly used, in order to obtain a highly oxidized sample, see reference [24] Reynosa-Martínez (2020). We have added a comment in lines 153-153 and a reference to address this comment.

  1. Line 119-120: Please be careful that Kim et al.[8] reported GO reduction by using H2SO4 and NaBH4 as an initial step, to compare their method (comment 1) with the previously reported method.

Response: The work of Kim et al. was already mentioned in the introduction (lines 50-55) but their work was further discussed in the revised version, lines 45-60.   

  1. In table1, How is C/O calculated?

Response: The carbon-oxygen ratio (C/O) was calculated from the atomic percentages obtained from the complete XPS scan. The atomic percentages of carbon and oxygen have been added in Table 1.

  1. Line 195-200 and Figure S5: 1) in Figure S5, please show the mechanism on a graphene structure. 2) What does it mean: “It is suggested that the -OH is protonated by an H+ from H2SO4, which occurs due to the electrostatic interaction between the negative charge of oxygen in H2SO4 and the positive charge of H in the -OH functional group.” From the basic chemistry, protonation of -OH occurs due to the negative charge of oxygen in –OH and positively charged H produced by acid. Isn’t it? 3) Please replace “Paso 3” with “step 3”. Furthermore, step 3 is not discussed in the text. This mechanism is the authors’ hypothesis or it is reported in other references. Can vinyl groups be produced by adding just sulphuric acid? Please discuss it.  

Response: The structure of graphene oxide was added to the proposed mechanism (now Figure S6), in order to better represent the effect of sulfuric acid on its functional groups. The complete discussion of the reaction mechanism, lines 237 to 243, was also added and the error related to the charges of the hydroxyl group and the acid proton, in lines 238 to 240, was corrected.

It has been reported that this acid is capable of dehydrating graphene oxide, through the protonation of the hydroxyl group and the subsequent release of the water molecule, see for example lines 53-55, reference [8] Kim (2012) in the manuscript. In this work, it is proposed that, after the release of the water molecule, a carbocation is formed that stabilizes by forming a double bond with the carbon neighbor.

  1. Figure S6: please show the mechanism on a graphene structure.

Response: The graphene structure was added to all proposed mechanisms; Figures S6 and S7.

  1. Figure S5 and Figure S7: What is the main role of H2SO4: -OH -> = or –OH->=O? Please discuss it.

Response: The effect of sulfuric acid does not occur selectively as it might affect different functional groups as described in the manuscript in lines 248 to 251. At the basal plane, the groups that can be transformed are hydroxyl and epoxides, while at the edges it is hydroxyl since the carboxyl and carbonyls are more stable, reference [11] Su (2014).

  1. In reference 10 “J. Phys. Chem. C 2010, 114, 19885–19890”, an interesting mechanism for the possible reduction of GO by NaHSO3 is proposed. In this mechanism, SO3 is produced as one of the final products. What is the possible reduction mechanism with H2SO4? Could SO3 be produced as one final product? When yes, that can explain, why adding SO3 has a negative effect on your process.

Response: The SO3 group is not considered as a final product or by-product of the reaction, since sulfuric acid in an aqueous medium is capable of recovering the released proton in a kind of acid-base equilibrium therefore it would not become SO3, see for example [37] Shin (2019). Nevertheless, the reaction mechanism proposed has been removed from the main text and the supporting information to avoid confusion on the effect of acid on graphene oxide because it is no longer necessary in the revised version and following other reviewers’ comments.

Reviewer 3 Report

The paper is interesting and it is likely that the results of the experiments justify the conclusions however the data needs to be presented with more clarity.

The peak fitting in both the XPS and the Raman needs to be moderated and justified.

The peak assignments in the XPS need to be clarified and possibly the fitting parameters tabulated.

The wide range of peak positions and full widths at half maxima is unrealistic and appears to be the result of an automated process to obtain a good fit and "quantify" the assigned groups.

After more careful and scientifically based peak fitting I would expect this paper to be publishable but as it is would not be a reliable reference to interpret future similar data. 

Author Response

Dear Editor,

We wish to submit the revised version of our manuscript materials-997842 entitled "Controlled Reduction of Graphene Oxide Using Sulfuric Acid", for consideration to the journal of Materials. The corrections to the manuscript and response to the reviewers are listed below (all changes are highlighted in yellow in the manuscript):

REVIEWER #3:

The paper is interesting and it is likely that the results of the experiments justify the conclusions however the data needs to be presented with more clarity.

  1. The peak fitting in both the XPS and the Raman needs to be moderated and justified.

Response: The computer programs used, as well as the functions and background used for the adjustment of the bands in XPS and Raman, were added in the experimental section in lines 105 to 112 and 115-117.

  1. The peak assignments in the XPS need to be clarified and possibly the fitting parameters tabulated.

Response: For the analysis of the XPS data, the CasaXPS program was used with a Shirley-type background and the binding energy was calibrated using the carbon energy at 284 eV. The bands was fitted at 284.9 eV (C=C/C-C), 285.9 eV (C-OH), 286.9 eV (C-O-C), 288.2 eV (C=O), and 289.3 eV (COOH) according to a reference [18] Perrozzi (2014), suggested by one of the reviewers. A more detailed description in this regard was added in the article on lines 105 to 112.

  1. The wide range of peak positions and full widths at half maxima is unrealistic and appears to be the result of an automated process to obtain a good fit and "quantify" the assigned groups.

Response: As mentioned above, the XPS analysis was performed in the CasaXPS program using as a guide its manual, reference [17] N. Fairley, (2009). The XPS analysis was carried out again in order to avoid variations in the position of the bands and the FWHM, the percentage of carbon with sp2 and sp3 hybridization and the functional groups were also recalculated. The discussion of the results is presented in lines 139 to 163, and can also be seen in Figure 2 and Table 1.

Reviewer 4 Report

In this work, the authors studied the reduction of graphene oxide by treatment with sulfuric acid at increasing concentration and with the aid of sulfuric anhydride. According to the authors it is possible to have a selective reduction of oxygenated groups present on the graphene skeleton.

I find the article very imprecise and before it can be considered for publication the authors must modify some substantial parts of the article.

1) The abstract must be completely rewritten. First of all, the percentages of the different oxygen groups included in the abstract derive from an analysis of the XPS data, but the authors did not specify it. But the analysis of the values starts from erroneous considerations and the authors must correct all the part concerning the contributions of the XPS peaks, and consequently the abstract itself. See the related part later in the referee.

2) I advise the authors to use fewer abbreviations in the abstract, to explain for example the formula SO3 with the name of the corresponding anhydride or sulfur trioxide, to specify “the characteristic signal of carbon sp2 “ instead of simply "C".

3) The introduction is very sparse, the authors should add some other references on the state of the art, especially as regards the comparison with other methods of GO reduction, which are multiple and also very effective.

Since the article deals with a reduction of a chemical type, it would at least be appropriate to mention the different reductive environments already studied in the literature and their effect on the different oxygenated groups present on the GO. The authors included some articles in the references in which different reducing agents are used, but then did not describe in detail the conditions used. For example, when describing the previous work with sulfuric acid and sodium borohydride (page 2, lines 43-44), actually the process followed by Kim et al., see Ref 8, involves a two-step reduction, including deoxygenation with sodium borohydride and subsequent dehydration with concentrated sulfuric acid. Authors are requested to improve the introduction and to be more detailed on the state of the art.

4) Authors must specify that the mixture of sulfuric acid and sulfur trioxide is commonly referred to as fuming sulfuric acid the first time they describe it, and then use this term in the following text.

5) The authors, in the paragraph concerning GO synthesis (page 2, line 58), write that they have added deionized water to the reaction mixture until a pH of 1 is obtained. I find it very difficult to reach a pH of 1 just by adding deionized water to a solution of concentrated sulfuric and phosphoric acids. The authors certainly have an extremely acid solution, still full of the ions of the starting reagents and washing with still concentrated HCl, water and ethanol cannot have completely removed them. Was a pH check of the final solution done?

6) Furthermore, the reduction of GO occurs at 90° C for 24 hours, with sulfuric acid at reflux. There is no test in which the GO is kept at 90 ° C for 24 hours under reflux in the absence of sulfuric acid, to verify the effect of temperature on the reduction of GO. The authors must add the data relating to the material treated at 90 °C at reflux in water for 24 and verify that the reduction has not occurred.

7) In the analysis of IR peaks, the authors erroneously assign the peak at 1200 cm-1 to COOH. It would be necessary to better verify which characteristic signal corresponds to the peak at 1200 cm-1, by reading carefully the literature. Do the authors know the C-O-C peak assignment?

8) The IR figure is not very legible, it would be appropriate to insert a widening of the spectra between 1750 and 1000 cm-1 to better describe the reduction of some peaks as the concentration of sulfuric acid increases. Please insert a zoom of the spectrum.

9) The bibliographic reference in the description of the IR is incorrect (ref 17), check the bibliography.

10) This is the weakest point of the work. The assignment and convolution of the peaks relative to carbon 1s is completely wrong. Please consider what is written in the literature, read the following bibliographical references and update the bibliography. After changing the energy values related to the different carbon contributions, the authors have to completely rewrite this part.

See for example: (a) Carbon, 77, 2014, 473-480 and (b) Nanotechnology, 22(5), 2011, 055705 (for hydrazine reduction) and add to the bibliography.

C 1s spectrum must be fitted summing the components related to the following carbon functional groups: C=C/C-C (C-H) (284.6-284.9 eV) related to aromatic ring (sp2) and hydrogenated carbon; C-OH (285.9 eV) related to hydroxyl groups, C–O–C (286.9 eV) related to epoxy groups, C=O (288.2

eV), C=O(OH) (289.3 eV), and the shake-up satellite (π–π∗ transition, 290.6 eV).

It is not possible to have such a high contribution of C sp3 in a graphenic material. Please correct.

Please correct also the abstract.

Rewrite Table 1 with the correct percentages.

10) On page 7, lines 189-190, the authors talk about FWHM, and then write percentages without specifying which of the peaks of the two bands they refer to. Please correct.

11) In Figure S5, where the GO reduction mechanism is hypothesized, the term deprotonation is incorrectly inserted instead of protonation (step 1).

12) In the main text the proposed reaction mechanism for OH groups reduction in GO (without the presence of SO3) is not well described, as in the SI. Please add the protonation, water leaving and carbocation generation and deprotonation steps.

I do not find it correct to hypothesize the mechanism on a methyl alcohol, better to leave the substituents undefined (also because they are benzene carbons) also for the de-epoxidation (Figure S6).

13) Check the correct indication of the compounds obtained by attachment of sulfuric acid to the hydroxyl group (Figure S7). It would be more correct to use the term phosphates or sulfuric acid esters, not simply the term ester (page 8, line 208) in the text.

14) What do the authors mean by the adjective "promissory" in the conclusions? did they mean “promising”? The conclusions are again a little too synthetic, they can be rewritten in more depth.

Author Response

Dear Editor,

We wish to submit the revised version of our manuscript materials-997842 entitled "Controlled Reduction of Graphene Oxide Using Sulfuric Acid", for consideration to the journal of Materials. The corrections to the manuscript and response to the reviewers are listed below (all changes are highlighted in yellow in the manuscript):

REVIEWER #4:

In this work, the authors studied the reduction of graphene oxide by treatment with sulfuric acid at increasing concentration and with the aid of sulfuric anhydride. According to the authors, it is possible to have a selective reduction of oxygenated groups present on the graphene skeleton.

  1. The abstract must be completely rewritten. First of all, the percentages of the different oxygen groups included in the abstract derive from an analysis of the XPS data, but the authors did not specify it. But the analysis of the values starts from erroneous considerations and the authors must correct all the parts concerning the contributions of the XPS peaks, and consequently the abstract itself. See the related part later in the referee.

Response: The abstract was modified to mention the use of XPS to obtain the values mentioned and further data was included.

  1. I advise the authors to use fewer abbreviations in the abstract, to explain for example the formula SO3 with the name of the corresponding anhydride or sulfur trioxide, to specify “the characteristic signal of carbon sp2” instead of simply "C".

Response: We reduced the use of abbreviations within the abstract, however, chemical formulas were kept to simplify the abstract.

  1. The introduction is very sparse, the authors should add some other references on the state of the art, especially as regards the comparison with other methods of GO reduction, which are multiple and also very effective.

Response: The introduction section was improved in order to enhance the contributions of this work by comparing it with other reduction methods. The modifications can be seen in lines 44-60

  1. Since the article deals with a reduction of a chemical type, it would at least be appropriate to mention the different reductive environments already studied in the literature and their effect on the different oxygenated groups present on the GO. The authors included some articles in the references in which different reducing agents are used, but then did not describe in detail the conditions used. For example, when describing the previous work with sulfuric acid and sodium borohydride (page 2, lines 43-44), actually the process followed by Kim et al., see Ref 8, involves a two-step reduction, including deoxygenation with sodium borohydride and subsequent dehydration with concentrated sulfuric acid. The authors are requested to improve the introduction and to be more detailed on the state of the art.

Response: The introduction was modified in order to explain in detailed the effect of each sulfur-compound (lines 45-50) on the oxygenated functional groups and thus compare them with the results of these work later (lines 154-157).

  1. Authors must specify that the mixture of sulfuric acid and sulfur trioxide is commonly referred to as fuming sulfuric acid the first time they describe it and then use this term in the following text.

Response: The clarification concerning the fuming sulfuric acid was added in the introduction section, lines 62-63, and the experimental section, in lines 88-89.

  1. The authors, in the paragraph concerning GO synthesis (page 2, line 58), write that they have added deionized water to the reaction mixture until a pH of 1 is obtained. I find it very difficult to reach a pH of 1 just by adding deionized water to a solution of concentrated sulfuric and phosphoric acids. The authors certainly have an extremely acid solution, still full of the ions of the starting reagents, and washing with still concentrated HCl, water, and ethanol cannot have completely removed them. Was a pH check of the final solution done?

Response: The step described is common in the preparation of graphene oxide. The pH of a suspension of GO in deionized water generally is between 2 and 3; measured using an OHAUS potentiometer, model starter 2100. This information was added in lines 84-86.

  1. Furthermore, the reduction of GO occurs at 90° C for 24 hours, with sulfuric acid at reflux. There is no test in which the GO is kept at 90 ° C for 24 hours under reflux in the absence of sulfuric acid, to verify the effect of temperature on the reduction of GO. The authors must add the data relating to the material treated at 90 °C at reflux in water for 24 and verify that the reduction has not occurred.

Response: The thermal reduction of graphene oxide occurred at temperatures above 200 °C. Furthermore, the reduction in water at low temperature does occur but with the aid of UV irradiation as it is shown in our manuscript and other works reports, see for example [13] Gallegos- Pérez (2020). These references and explanation were included in the text. Although these experiments we considered are unnecessary given the known behavior of graphene oxide in the conditions described by the reviewer, at the moment it would be unfeasible to perform them due to the current worldwide pandemic.

  1. In the analysis of IR peaks, the authors erroneously assign the peak at 1200 cm-1 to COOH. It would be necessary to better verify which characteristic signal corresponds to the peak at 1200 cm-1, by reading carefully the literature. Do the authors know the C-O-C peak assignment?

Response: The bands at 1040 and 1200 cm-1 both correspond to C-O bonds, however, similar analysis on GO have assigned the band at 1040 cm-1 to the presence of the epoxide group, whereas, the band at 1200 cm-1 to the carboxylic acid. See for example references [23] Seredych (2011). The text was modified to specify that the bands both correspond to the C-O bond.

  1. The IR figure is not very legible, it would be appropriate to insert a widening of the spectra between 1750 and 1000 cm-1 to better describe the reduction of some peaks as the concentration of sulfuric acid increases. Please insert a zoom of the spectrum.

Response: An image was added to Figure 1 in which the FT-IR spectra are observed from 2000 to 1000 cm-1 to better appreciate the effect of sulfuric acid concentration on oxygenated functional groups of GO.

  1. The bibliographic reference in the description of the IR is incorrect (ref 17), check the bibliography.

Response: References were checked and this error was corrected.

  1. This is the weakest point of the work. The assignment and convolution of the peaks relative to carbon 1s is completely wrong. Please consider what is written in the literature, read the following bibliographical references and update the bibliography. After changing the energy values related to the different carbon contributions, the authors have to completely rewrite this part.

See for example: (a) Carbon, 77, 2014, 473-480 and (b) Nanotechnology, 22(5), 2011, 055705 (for hydrazine reduction) and add to the bibliography.

C 1s spectrum must be fitted summing the components related to the following carbon functional groups: C=C/C-C (C-H) (284.6-284.9 eV) related to aromatic ring (sp2) and hydrogenated carbon; C-OH (285.9 eV) related to hydroxyl groups, C–O–C (286.9 eV) related to epoxy groups, C=O (288.2 eV), C=O(OH) (289.3 eV), and the shake-up satellite (π–π∗ transition, 290.6 eV).

Response: The XPS analysis was performed again, this time using the position of the bands suggested by the reviewer, reference [18] Perrozzi (2014). The deviation in the position of the bands was also calculated to avoid significant variations and the percentages for each link were also recalculated. A more detailed description in this regard was added in the article in lines 108 to 115 in the experimental section and the discussion of the results is presented in lines 144 to 168 and Table 1, the graphic representation is presented in Figure 2.

  1. It is not possible to have such a high contribution of C sp3 in a graphenic material. Please correct.

Response: As already mentioned, the XPS analysis was performed again, so the percentages of carbon and functional groups were corrected (Table 1), now it shows more clearly the effect of the acid on the graphene oxide.

  1. Please correct also the abstract.

Response: The abstract was rewritten according to the modification performed at the XPS analysis.

  1. Rewrite Table 1 with the correct percentages.

Response: Table 1 was rewritten according to the modification performed at the XPS analysis.

  1. On page 7, lines 189-190, the authors talk about FWHM, and then write percentages without specifying which of the peaks of the two bands they refer to. Please correct.

Response: It was specified that it was referring to the FWHM values of the G band (line 232).

  1. In Figure S5, where the GO reduction mechanism is hypothesized, the term deprotonation is incorrectly inserted instead of protonation (step 1).

Response: This error was corrected in what is now Figure S6.

  1. In the main text the proposed reaction mechanism for OH groups reduction in GO (without the presence of SO3) is not well described, as in the SI. Please add the protonation, water leaving and carbocation generation and deprotonation steps.

Response: Added step three to the description of the reaction mechanism for dehydroxylation at lines 241-242.

  1. I do not find it correct to hypothesize the mechanism on a methyl alcohol, better to leave the substituents undefined (also because they are benzene carbons) also for the de-epoxidation (Figure S6).

Response: The graphene oxide structure was added to all the reaction mechanisms to clarify the effect of acid on its functional groups, it can be seen in Figure S7.

  1. Check the correct indication of the compounds obtained by attachment of sulfuric acid to the hydroxyl group (Figure S7). It would be more correct to use the term phosphates or sulfuric acid esters, not simply the term ester (page 8, line 208) in the text.

Response: The reaction mechanism proposed has been removed from the main text and the supporting information to avoid confusion on the effect of acid on graphene oxide because it is no longer necessary based on the new results.

  1. What do the authors mean by the adjective "promissory" in the conclusions? did they mean “promising”? The conclusions are again a little too synthetic, they can be rewritten in more depth

Response: The conclusion was rewritten to make it more complete and the word "promissory" was modified.

Round 2

Reviewer 2 Report

In my opinion, the manuscript has been improved and can be accepted in present format.

Author Response

Thanks for your previous contributions

Reviewer 3 Report

The authors response demonstrates the XPS fitting still requires some additional clarification though the references and description of how it was achieved is an improvement. There are some contradictions between the text and what the fitted spectra show and the percentage to each chemical state cannot be as accurate as implied. The binding energy assignments are acceptable but the FWHM of the fitted peaks is not fixed at 1eV as suggested in the text below. You would expect some variation as peaks are fitted but not as much as the figures show.  

The processing of the XPS data of the C1s region was carried out in the CasaXPS program using
a Shirley background [17]. Carbon with sp2 and sp 106 3hybridization (C-C/C=C) was assigned to the
107 binding energy of 284.9 eV and adjusted with a Gaussian function with 30% of a Lorentzian function
108 with symmetric shape [17, 18]. The signals corresponding to the oxygenated functional groups were
109 assigned to 285.9 eV (C-OH), 286.9 eV (C-O-C), 288.2 eV (C=O), and 289.3 eV (COOH) [12, 18–21]
110 which were also fitted with a Gaussian with 30% Lorentzian function and with symmetric shape, and
111 for all cases, the full width half maximum (FWHM) was kept at 1 eV and The binding energy was
112 calibrated using the carbon energy at 284 eV [17].

Author Response

Editor

Materials

Dear Editor,

We wish to submit the revised version of our manuscript materials-997842 entitled "Controlled Reduction of Graphene Oxide Using Sulfuric Acid", for consideration to the journal of Materials. The corrections to the manuscript and response to the reviewers are listed below (all changes are highlighted in yellow in the manuscript):

Reviewer #3

  1. The author’s response demonstrates the XPS fitting still requires some additional clarification though the references and description of how it was achieved is an improvement. There are some contradictions between the text and what the fitted spectra show and the percentage to each chemical state cannot be as accurate as implied. The binding energy assignments are acceptable but the FWHM of the fitted peaks is not fixed at 1eV as suggested in the text below. You would expect some variation as peaks are fitted but not as much as the figures show.

“The processing of the XPS data of the C1s region was carried out in the CasaXPS program using a Shirley background [17]. Carbon with sp2 and sp3 hybridization (C-C/C=C) was assigned to the binding energy of 284.9 eV and adjusted with a Gaussian function with 30% of a Lorentzian function with symmetric shape [17, 18]. The signals corresponding to the oxygenated functional groups were assigned to 285.9 eV (C-OH), 286.9 eV (C-O-C), 288.2 eV (C=O), and 289.3 eV (COOH) [12, 18–21] which were also fitted with a Gaussian with 30% Lorentzian function and with symmetric shape, and for all cases, the full width half maximum (FWHM) was kept at 1 eV and the binding energy was calibrated using the carbon energy at 284 eV [17].”

Response: The reviewer is correct, the FWHM of the signals was not kept at 1 eV. The 1eV value is the set point at the start of the deconvolution using the CasaXPS program; we have removed that incorrect statement. After fitting each bond, the FWHM value increases or decreases as the function adjusts to the spectrum and, for this specific case, the change in the FWHM is attributed to a change in the number of bonds in the functional group. See for example [17] Fairley (2009), already included in the text.

Reviewer 4 Report

The authors have made the necessary corrections required in the first revision, responding to almost all my requests and modified the article making it certainly more readable and rigorous. There remains only one fix that I suggest to the authors before I can publish this version.

In lines 124-126 the authors continue to assert that the: “two bands around 1040 and 1,200 cm-1 were observed corresponding to C-O bonds usually associated with a carboxylic acid (COOH) and the presence of the epoxide (C-O-C) groups, respectively.

The signal at 1040 cm-1 is due to C-O signal from hydroxyl, while at 1200 cm-1 the C-O-C signal characteristic of epoxides. The C-O(=O) are at about 1420 cm-1.

Please correct it.

After the correction the article is suitable for publication.

Author Response

Editor

Materials

Dear Editor,

We wish to submit the revised version of our manuscript materials-997842 entitled "Controlled Reduction of Graphene Oxide Using Sulfuric Acid", for consideration to the journal of Materials. The corrections to the manuscript and response to the reviewers are listed below (all changes are highlighted in yellow in the manuscript):

Reviewer #4

  1. The authors have made the necessary corrections required in the first revision, responding to almost all my requests, and modified the article making it certainly more readable and rigorous. There remains only one fix that I suggest to the authors before I can publish this version.

In lines 124-126 the authors continue to assert that the: “two bands around 1040 and 1,200 cm-1 were observed corresponding to C-O bonds usually associated with a carboxylic acid (COOH) and the presence of the epoxide (C-O-C) groups, respectively.”

The signal at 1040 cm-1 is due to C-O signal from hydroxyl, while at 1200 cm-1 the C-O-C signal characteristic of epoxides. The C-O(=O) is at about 1420 cm-1.

Please correct it.

After the correction, the article is suitable for publication.

Response: The assignment of bands in the FTIR has been corrected in lines 124-125.
